# DeepBiomarker2: Prediction of Alcohol and Substance Use Disorder Risk in Post-Traumatic Stress Disorder Patients Using Electronic Medical Records and Multiple Social Determinants of Health

**DOI:** 10.3390/jpm14010094

**Published:** 2024-01-14

**Authors:** Oshin Miranda, Peihao Fan, Xiguang Qi, Haohan Wang, M. Daniel Brannock, Thomas R. Kosten, Neal David Ryan, Levent Kirisci, Lirong Wang

**Affiliations:** 1Computational Chemical Genomics Screening Center, Department of Pharmaceutical Sciences/School of Pharmacy, University of Pittsburgh, Pittsburgh, PA 15213, USA; osm7@pitt.edu (O.M.); pef14@pitt.edu (P.F.); xiq24@pitt.edu (X.Q.); 2School of Information Sciences, University of Illinois Urbana-Champaign, Champaign, IL 61820, USA; haohanw@illinois.edu; 3RTI International, Durham, NC 27709, USA; mbrannock@rti.org; 4Menninger Department of Psychiatry, Baylor College of Medicine, Houston, TX 77030, USA; kosten@bcm.edu; 5Department of Psychiatry, School of Medicine, University of Pittsburgh, Pittsburgh, PA 15213, USA; nryan@pitt.edu; 6Center for Education and Drug Abuse Research, Department of Pharmaceutical Sciences/School of Pharmacy, University of Pittsburgh, Pittsburgh, PA 15213, USA; levent@pitt.edu

**Keywords:** post-traumatic stress disorder, alcohol, and substance use disorder, social determinants of health, psychotherapy, natural language processing, deep learning, biomarker identification

## Abstract

Prediction of high-risk events amongst patients with mental disorders is critical for personalized interventions. We developed DeepBiomarker2 by leveraging deep learning and natural language processing to analyze lab tests, medication use, diagnosis, social determinants of health (SDoH) parameters, and psychotherapy for outcome prediction. To increase the model’s interpretability, we further refined our contribution analysis to identify key features by scaling with a factor from a reference feature. We applied DeepBiomarker2 to analyze the EMR data of 38,807 patients from the University of Pittsburgh Medical Center diagnosed with post-traumatic stress disorder (PTSD) to determine their risk of developing alcohol and substance use disorder (ASUD). DeepBiomarker2 predicted whether a PTSD patient would have a diagnosis of ASUD within the following 3 months with an average c-statistic (receiver operating characteristic AUC) of 0.93 and average F1 score, precision, and recall of 0.880, 0.895, and 0.866 in the test sets, respectively. Our study found that the medications clindamycin, enalapril, penicillin, valacyclovir, Xarelto/rivaroxaban, moxifloxacin, and atropine and the SDoH parameters access to psychotherapy, living in zip codes with a high normalized vegetative index, Gini index, and low-income segregation may have potential to reduce the risk of ASUDs in PTSD. In conclusion, the integration of SDoH information, coupled with the refined feature contribution analysis, empowers DeepBiomarker2 to accurately predict ASUD risk. Moreover, the model can further identify potential indicators of increased risk along with medications with beneficial effects.

## 1. Introduction

Posttraumatic stress disorder (PTSD) and alcohol and substance use disorder (ASUD) often co-occur, with an estimated prevalence of ASUD amongst individuals with PTSD of 46% in the United States alone [1]. Patients with ASUD experience higher rates of PTSD, with the highest rates reported in patients with both alcohol and drug use disorders [2]. The current literature has enlisted mechanisms that may explain this co-occurrence: (1) ASUD could exacerbate the risk of developing PTSD, as patients tend to lead a high-risk lifestyle, which increases the chances of being exposed to or experiencing a traumatic event (e.g., sexual assault under the influence of substances) [3]; (2) PTSD development can precede ASUD as patients use substances to self-medicate their PTSD symptoms [4]; (3) genetic influences on the onset, maintenance, or etiology of both disorders; and (4) multiple aspects of well-being such as psychosocial risk and protective factors extracted from social determinants of health (SDOH) could be related to a shared underlying factor affecting the overall quality of life. While various studies suggest that both PTSD and ASUD share common dysfunctions in numerous biological systems, it is paramount to determine multiple indicators of increased and decreased risk that are responsible for the development of ASUD in PTSD, and this can be achieved via the application of novel analytic technologies to data-mine Electronic medical record (EMR) data from these patients. In addition, the current treatment of PTSD and ASUD is limited [5].

Social determinants of health (SDoH) are “conditions or environments in which people are born, grow, live, work, and age” [6]. Five key SDoH domains that have significant impacts on human health are (1) economic stability, (2) education, (3) health and healthcare, (4) neighborhood and the built environment, and (5) the social and community context [7]. These complex, integrated, and overlapping social and economic systems are, in turn, responsible for most health inequalities and poor health outcomes existing today. EMRs are an important component of clinical practice and documentation. However, a major limitation of EMRs is the lack of reliable SDoH information and documentation, which is strongly associated with mental health [8]. EMRs collect clinical information such as diagnosis, medication use, laboratory test results, vital signs, procedures, and other data in a systemic fashion. Other nonclinical determinants of health such as age, race, and ethnicity are collected in a structured EMR format. Current studies have linked a variety of SDoH parameters such as neighborhood socioeconomic status (nSES) indicators with disease risk factors to improve the accuracy of risk prediction models [9,10]. By extrapolating information obtained from both conventional (e.g., EMR sources) and non-conventional sources (e.g., SDoH databases, clinical EMR notes, census data), one can use these “big data” for risk prediction and the development of interventions to improve multiple clinical outcomes, especially focusing on high-risk patients such as patients with multiple comorbidities. However, to the best of our knowledge, few studies have assessed the importance of this wide range of multimodal information, which includes diagnosis, medication use, laboratory test results, individual-level SDoH indicators (e.g., race, age, gender, etc.), neighborhood-level SDoH indicators (e.g., nSES index, etc.), and psychotherapy status in the prediction of outcomes of patients, especially with mental disorders.

Deep learning/data mining algorithms translate “big data” into valuable information for hypothesis generation through deep hierarchical feature construction to capture long-range dependencies in the EMR data [11]. Deep learning techniques learn certain features directly from the data itself, without any human guidance, thus allowing for the automatic discovery of latent data relationships that might otherwise be hidden. Various deep learning models offer unique advantages in addressing challenges associated with temporal EHR data by using the multilayer perceptron (MLP) [12], the restricted Boltzmann machine (RBM) [13], a convolutional neural network (CNN) [14], a recurrent neural network (RNN) [15,16], and transformer-based architectures [17], to name a few. A systematic review found RNNs, including Long short-term memory (LSTM) and the Gated Recurrent Unit (GRU), effectively handle sequential information in multiple studies, while CNNs capture spatial correlations in fewer studies [18]. Additionally, the complex structure of temporal EHR data poses difficulties in standard learning algorithms, and four major challenges are currently identified: data irregularity, sparsity, heterogeneity, and model opacity. Intrinsically interpretable models, like the Risk-calibrated Supersparse Linear Integer Model (RiskSLIM) [19] and AutoScore [20], integrate machine learning with deep learning for practical clinical score generation. In summary, these models contribute diverse strengths, encompassing data augmentation, transferability, interpretability, and effective handling of sequential and spatial information in temporal EHR data. Deep learning models still encounter other challenges related to their interpretability and their capacity to offer uncertain estimates, which is less than ideal for clinical applications [18,21,22,23]. Future studies are necessary to explore the development of more comprehensive and integrated solutions to enhance the effectiveness of handling temporal EHR data. Researchers are encouraged to integrate clinical domain knowledge into their study designs and focus on improving model interpretability, ultimately facilitating more seamless clinical implementation [18,24]. Clinical domain knowledge encompasses expertise in clinical medicine and healthcare, covering medical practices, diseases, patient care, and treatment protocols. Examples include disease pathophysiology, treatment guidelines, patient care protocols, medical terminologies, diagnostic criteria, pharmacology, epidemiology, patient history analysis, clinical research methods, and familiarity with healthcare workflows. Integrating clinical domain knowledge is crucial for designing effective healthcare solutions, interpreting medical data, and ensuring artificial intelligence models align with real-world clinical scenarios [24,25,26,27,28,29,30,31].

By analyzing the EMR of PTSD patients and leveraging clinical domain knowledge, we can find indicators of increased risk or medications that may have the potential to reshape the trajectory of disease progression. We are especially interested in finding medications associated with a high risk or a low risk of developing adverse outcomes such as developing substance abuse disorder among PTSD patients. This, in turn, can be used to design better treatment options for those patients. In this study, we improved our previous deep-learning-based Deepbiomarker to develop our latest version, DeepBiomarker2. We refined the relative contribution analysis for the identification of important features, used natural language processing to extract psychotherapy from clinical notes, and integrated multiple SDOH and psychotherapy parameters in our model. We then applied DeepBiomarker2 to ASUD risk prediction, provided refined results specific to high-risk cohorts, proposed new interdisciplinary hypotheses, and identified risk/protective medications for the prevention of PTSD developing into ASUD. In the development of DeepBiomarker2, clinical domain knowledge played a pivotal role. We enhanced model applicability and interpretability by incorporating clinical domain knowledge through feature selection, collaboration with healthcare professionals, and transparent reporting to contribute to model relevance and practical implementation in healthcare settings. Collectively, we anticipate that health professionals can improve the classification of patients based on their complexity and heterogenicity, develop targeted interventions for better health outcomes, and reduce existing health disparities at lower costs.

## 2. Materials and Methods

### 2.1. Data Source

We included data from January 2004 to October 2020 from the Neptune system at the University of Pittsburgh Medical Center (UPMC), which houses EMR from the UPMC health system for research purposes (rio.pitt.edu/services). The database includes multimodal information: demographic information, diagnoses, encounters, medication prescriptions, prescription fill history, and laboratory tests. We used ASUDs after the diagnosis of PTSD. PTSD and ASUD patients were identified by ICD9/10 codes (‘309.81’, ‘F43.10’, ‘F43.11’, and ‘F43.12’) and ICD9/10 codes (See Appendix B), respectively. The medication fills include medications that a patient had filled at commercial pharmacies, collected, and reported by the clearing house SureScripts.

### 2.2. Data Preparation

Data preparation was performed similarly to that described in our previous paper DeepBiomarker [32]: For a given PTSD patient without a previous diagnosis of ASUD at an index date, our primary aim was to predict whether the patient will experience ASUDs within the next 3 months. In our study, a case was defined as a PTSD patient who had a record of ASUD within the next 3 months, while a control was defined as a PTSD patient with no record of ASUD in the next 3 months after the index date. The index date can be any encounter date after the PTSD diagnosis but before the first diagnosis of ASUD. If a patient had multiple encounters satisfying the inclusion criteria, we only considered the latest encounter to mimic the latest status of the patient. We excluded patients who had a diagnosis of PTSD and ASUD on the same day or experienced ASUD before the PTSD diagnosis. The patient was also required to have no record of ASUD within one year before the index date to negate the possibility of a previous history of ASUD (Appendix A). The reason for using a 3-month time window for prediction and a one-year observation time window for data collection is that we are interested in risk prediction of the near future (e.g., within 3 months) and the effects of medication use on patient outcomes. We assume that the effects of a medication cannot last for too long (e.g., no more than 1 year). We used multimodal information such as diagnosis, medications, and lab test results 1 year preceding the index date, as well as SDoH and psychotherapy information at the index date as the input. We specifically included lab test results that had high frequencies and were coded as abnormal by searching results that were labeled as “ABNORMAL”, “HIGH”, or “LOW”. We did not consider lab test results with a low frequency (less than 100 times). We clustered diagnosis codes into diagnosis groups based on the first 3 letters of their ICD-10 codes. Medication names were converted to their respective unique DrugBank IDs. Lastly, for each encounter, the associated multimodal information, namely diagnosis, medication, and lab tests, was packed into a sequence based on their respective disease categories, DrugBank IDs, and lab test IDs.

### 2.3. Dataset Splitting

We split our dataset with a ratio of 8:1:1, where 8 subsets were used as the training dataset, one subset was used as the validation dataset to find the optimal parameters, and the last subset was used as the test set to evaluate the generalization of our model.

### 2.4. SDoH Data

For each PTSD patient, we included both the individual-level SDoH and neighborhood-level SDoH data. Individual-level SDoH features such as race, age, and gender were extracted from the demographic information in EMR and coded similarly to diagnosis codes to input in the models. We also used neighborhood-level SDoH features (see Appendix B) such as racial segregation, neighborhood socio-economic status, percentage of non-citizens, a person of color index, the normalized difference vegetation index, the aridity index, percentage of male widowers, percentage of US citizens, percentage of households with limited English proficiency, income segregation, percentage of population with same-sex marriage, the urban index, percentage of population who are separated from their partner, and percentage of households with transportation barriers, which were calculated using their respective formulas and extracted from the American community survey (ACS) (See Appendix A about SDoH parameters). Neighborhood-level SDoH factors are geographically derived neighborhood-level SDoH parameters that can be used for the assessment of healthcare utilization [33]. The ACS is a rolling survey of the US population that gathers information, such as ancestry, educational level, income level, language proficiency, migration status, disability status, employment status, and housing characteristics, across 1298 variables [34]. The ACS releases estimates at the regional, state, and county levels every year, and data at the census tract, block group, and zip code levels are available every 5 years. Structural racism demands a multidimensional measure to address racial and income disparities [35]. Previous studies have used all of the above indexes including an index of concentration at the extremes (ICE) to represent a geographical area–based measure of the socioeconomic deprivation experienced according to their neighborhood [36]. Although the efficacy of these SDoH parameters such as ICE is established at the census tract level, its utility at the zip code level, where it offers potentially more stable estimates of disparities, remains relatively unexplored [37,38]. For our study, a patient’s 5-digit zip code at the index date was used. Collectively, the SDoH data were mapped and later used as input in our model.

### 2.5. Trauma-Focused Psychotherapy and Cognitive Behavioral Therapy Data

For each PTSD patient, we extracted both cognitive behavioral therapy (CBT) and trauma-focused psychotherapy data. Trauma-focused psychotherapy included cognitive processing therapy (CPT), prolonged exposure (PE), and eye movement desensitization and reprocessing (EMDR), which were extracted from respective clinical notes in the EMR using natural language processing techniques. The iterative extraction process was performed as follows (Appendix A): we created a custom keyword dictionary with the help of subject matter experts and extracted only those sentences that included the keywords or related keywords that had: “cognitive processing therapy”, “prolonged exposure”, “eye movement desensitization and reprocessing”, and “cognitive behavioral therapy”. We then created a sentence dictionary that would identify only those patients who have an active status. We then used a pre-trained model all-mpnet-base-v2, a sentence transformer-based natural language model. This model is based on the MPNet architecture and has the highest performance in generating sentence embeddings [39]. We used the original form, where the output dimension was 768. We then identified which patients underwent or are undergoing CBT and trauma-focused psychotherapy, and we coded it similar to SDoH to input in the models.

### 2.6. DeepBiomarker2

We leveraged the Pytorch_EHR framework developed by Rasmy et al. as the foundation of our analysis. These deep learning models based on multiple recurrent neural networks were used to analyze and predict clinical outcomes [40]. We implemented two models: TLSTM (Time-aware long short-term memory) and RETAIN (Reverse Time Attention Model). TLSTM is a specialized neural network architecture designed for modeling time-series data with temporal dependencies. Unlike traditional LSTMs, they consider timestamps, enabling them to capture the data’s temporal context during training and prediction. These models are valuable in handling irregularly sampled time-series data with varying intervals and missing data. It decomposes memory cells, integrates elapsed time information, and effectively captures temporal dynamics, enhancing its utility for tasks involving irregularly timed data sequences [41]. RETAIN (Reverse Time Attention Model) is a specialized neural network architecture that leverages both patient-level and visit-level information to make predictions about patient outcomes using EMR data. The model incorporates attention mechanisms, which enable it to weigh the significance of different medical visits and patient history, providing interpretable insights into why specific predictions are made. In conjunction with these algorithms and our previous model DeepBiomarker, we further modified the framework and named it DeepBiomarker2 (Figure 1). This enhanced model represented a significant advancement regarding (a) the extraction of psychotherapy status (both cognitive behavioral status and trauma-focused psychotherapy) from clinical notes, (b) the integration of individual lab tests, SDoH parameters, psychotherapy status, medications, and diagnosis as the input, so that we can assess the important clinical and non-clinical factors associated with ASUD risk, and (c) the refinement of the contribution analysis module, a pivotal component of our approach. This refinement involved optimizing the relative contribution analysis, a method used to quantify the impact of key factors. We assessed these factors by measuring the observed changes in the model’s predictions. In our study, we followed the same parameters as our previous versions to maintain consistency and ensure methodological robustness and comparability. Embedding dimension 128 determined the dimensionality of embedded data representations, impacting the model’s ability to capture complex relationships. The hidden size of 128 specified the number of neurons in each recurrent layer, influencing the model’s capacity to capture intricate data patterns. A dropout rate of 0.2 served as a regularization measure during training, preventing overfitting by randomly deactivating neurons. With eight layers, our model delved deeply into data hierarchies, although at the cost of increased complexity. An input size of 30,000 signified the breadth of features considered, while a patience value of 3 dictated early stopping based on validation performance. Together, these parameters underpinned our deep learning model’s architecture, ensuring reliable and consistent analysis of factors associated with ASUD risk among PTSD patients. To estimate the standard deviations of the accuracy, we repeated our calculations five times for each of the algorithms.

### 2.7. Statistical Analysis

Assessment of the importance of the clinical factors for predicting ASUD events.

To examine the importance of the clinical factors for the prediction of ASUDs, we calculated the relative contribution (*RC*) of each feature on the ASUD [42]. The *RC* of a feature was calculated as the median contribution of the feature to events divided by the median contributions of this feature to no events. The contributions were estimated by a perturbation-based approach [43]. The *RC* value and significance calculation are shown as follows where *FC* represents the feature contribution:$$RC\;value=\frac{{median(FC}_{with\;event})}{{median(FC}_{without\;event})}$$
$$RC\;significance=Wilcoxon\;rank\;sum\;test\;p\;value\;({FCs}_{with\;event}\;and\;{FCs}_{without\;event})$$

The *FC* value was the total value of the feature within the same patient if the feature appeared more than once in that patient. In most cases, *FCs* did not follow a normal distribution. As such, the medians of *FCs* were used in the *RC* calculation instead of the mean value, and the significance of *RCs* was represented by the *p*-value of the Wilcoxon rank sum test with the comparison of the median difference between *FCs* with and without events [44]. False discovery rate (FDR) adjustment was used to reduce the type I error caused by multiple comparisons. FDR is the expected ratio of the number of false positive results to the number of total positive test results [45]. The FDR adjustment was represented by an FDR-adjusted *p* value with alpha = 0.05.

We improved our assessment by normalizing the *FC* value and scaling the *RC* value for all our features. The improved *FC* formula is the ratio of the summary of the contribution of a feature and the summary of the contribution of all the features. Next, we performed scaling, where the *RC* for a PTSD diagnosis was scaled to 1 and the scale factor generated was applied to obtain the final *RC* value for each of the other features. *FC* normalization accounts for heterogeneity in the numbers of encounters across patients, which may otherwise inflate the contribution of features observed in patients with higher healthcare utilization, and by normalization, we estimate the contribution of a feature while considering all features from that patient/sample. The step of scaling is to distinguish beneficial factors from risk factors using a common internal factor as a reference. Because of the disparity in visits among cases and controls, even PTSD might have a beneficial or risk effect when only considering the raw *RCs*. We also would emphasize that this is still an active area in the model interpretation, and more complex and reasonable approaches might come out in the future. To evaluate the approach, we can use previous well-known knowledge to validate the performance of analysis, e.g., whether an approach can “discover” previous findings and reveal some new insights for hypothesis generations to generate new knowledge.

Assessment of model performance.

The model performance was evaluated by the area under the ROC curve (AUROC) in both the validation set and test set. The precision, recall, and F1 scores were calculated in the test sets. The mean and standard deviation of five repeats of those metrics were reported. We used both deep learning and logistic regression to compare the performance of our model. To assess the effect of SDoH, we repeated our analysis ten times, averaged the coefficient factors of SDoH parameters, and calculated the *p* values using a *t*-test.

## 3. Results

The performance of DeepBiomarker2 on the ASUD prediction.

We identified 38,807 PTSD patients from UPMC EMR data. We further identified 7927 cases and 7685 controls from patients with more than 1 year of EMRs before the diagnosis of PTSD (Appendix A). Those samples were split into an 8:1:1 ratio for training, validation, and test sets. The performance metrics of the DeepBiomarker2 can be found in Table 1.

As shown in Table 1, the deep learning models TLSTM and RETAIN algorithms implemented in DeepBiomarker2 all showed excellent performance on ASUD prediction, i.e., all yielded an AUC ≥ 0.90 (Appendix A). The performance of deep learning (AUC above 0.93) was better than LR (0.85). The performance of models with SDoH was slightly better than those without SDoH.

### 3.1. Important Indicators for the ASUD Prediction

As we described previously, we followed a perturbation-based estimation approach to calculate the relative contribution of each feature on the prediction of ASUD (Appendix A). Table 2, Table 3, Table 4 and Table 5 enlist the top important abnormal lab tests, medication use, diagnosis, and SDoH parameters, respectively. The most important indicators were ranked based on the highest number of cases and controls and the most significant *p*-values. We can see that in Table 2, HGB with RC of 1.45, along with other abnormal lab tests with *RC* > 1, are indicators of increased risk for ASUD. In Table 3, pain medications such as acetaminophen and oxycodone both have an *RC* of *RC* > 1, which are categorized as indicators of increased risk for ASUD while medications such as clindamycin and enalapril have an *RC* < 1 and are categorized as indicators of decreased risk for ASUD. In Table 4, diagnoses such as routine lab examinations are categorized as protective factors for ASUD (*RC* = 0.71) while other chronic pain is categorized as a risk factor for ASUD (*RC* = 1.17).

### 3.2. Overall Lab Test-Based Indicators of Comorbidities and Disease Burdens for ASUD Prediction

Through further analysis of the DeepBiomarker2 model, we identified the most important lab tests as the biomarkers (Appendix A). These biomarkers are strongly correlated to ASUD and PTSD, along with their implications for adjoining diagnosis and medication use. These lab tests are indicators of underlying comorbidities and thus can be considered measurements of disease burdens.

## 4. Discussion

We developed and applied our deep learning model DeepBiomarker2 to predict the risk of ASUDs in PTSD patients based on abnormal results of regular lab tests in the last year together with the diagnosis and medications used in the same period, as well as SDoH parameters. The model yielded very good performance with an AUC score above 0.93. The improvement might be due to the fact that DeepBiomarker2 can also consider the sequential information of these multimodal features. This study marks the first to establish a connection between these lab test results and the risk of ASUD among PTSD patients. However, further research investigating the relationship between these biomarkers and the risk of ASUD in PTSD patients is warranted, as there remains a gap in the existing literature on this topic. To further refine our understanding of specific biomarkers of PTSD and ASUD, we have categorized our top biomarkers as follows:

### 4.1. Biomarkers Closely Related to PTSD and ASUD

Inflammatory-based biomarkers. We have identified two inflammatory-based biomarkers that are potentially useful for assessing the risks of ASUDs in PTSD patients. Current research has emphasized the importance of incorporating inflammatory biomarkers in risk prediction models to further catapult mental disorder research efforts. Several studies showed PTSD patients had elevated levels of WBC and neutrophil levels in their system on account of the activation of multiple inflammatory pathways. Abnormal WBCs and neutrophil levels express tissue function and release pro-inflammatory and pro-coagulant molecules to promote thrombus formation, including platelet activation and adhesion, which potentially increase the risk of cardiovascular disease in these patients [46]. Another study found low WBC and neutrophil levels in alcoholics but higher WBC levels in cannabis, inhalants, tobacco, and opioid users, and no significant levels of WBCs amongst cocaine users. At the same time, low neutrophil levels were seen in cannabis users and high levels in inhalants and opioid users [47,48,49,50,51].

Heme-based biomarkers. The PTSD patient population might have pathologies that are related to hematopoiesis, inflammation, endothelial function, and coagulability, depending on abnormal levels seen in these patients [52]. Current epidemiological studies have shown RBCs may interact with the inflammatory system and platelets. Once exposed to oxidative stress, they acquire a senescent phenotype promoting a pro-inflammatory and pro-atherogenic state [53]. We found elevated levels of hemoglobin, red blood cells, hematocrit, RDW, MCH, and MCHC in depressed patients. We also found low hemoglobin levels in alcoholics, cannabis, and heroin users [54]. At the same time, other heme biomarkers are discussed in Appendix A. To the best of our knowledge, we would be the first to propose these heme biomarkers as possible biomarkers for the risk of ASUDs in PTSD patients.

Liver-based biomarkers. Albumin is a protein shown to possess free-radical scavenging properties that act as a selective antioxidant. There are studies examining the role of serum albumin levels in patients with psychiatric diseases [55,56]. The current research hypothesizes that low serum albumin levels in depressed patients might be due to the activation of inflammatory responses in these patients [57,58,59]. Another study found low serum albumin levels in drug addicts and higher serum albumin levels in alcoholics in an emergency department setting [60]. However, it is important to thoroughly examine the association between developing ASUD risk and albumin to demonstrate it as a prognostic biomarker amongst PTSD patients.

Additional biomarkers including calcium, chloride, carbon dioxide, total protein, protein in the urine, leukocyte esterase, and urea nitrogen were identified as factors associated with increased risk of ASUD in our cohort of patients with PTSD.

### 4.2. Effect of Medication Use on PTSD for ASUD Prediction

Indicators of increased risk:

Pain medications: PTSD patients prescribed pain medications such as acetaminophen, oxycodone, hydrocodone, and gabapentin have significantly higher PTSD symptom severity scores, with opiate analgesics use associated with the highest scores [61]. The rationale associated with the use of both opiate and non-opiate analgesics leads to the hypothesis suggesting the dysregulation of the opioid system in both PTSD and ASUD. While physical injury at the time of PTSD accounts for ongoing pain symptoms seen in these patients, there is a possibility that the emotional and social impact of these traumas may further exacerbate ASUD risks among these patients.

Alprazolam: Alprazolam is a medication that belongs to the class of benzodiazepines. Studies have found that there may be an increased incidence of PTSD in patients treated with alprazolam immediately after exposure both in the civilian and veteran populations [62]. Other studies found that alprazolam has been associated with disproportionate harm compared to other benzodiazepines, especially among people in opioid substitution treatment, and drug-related deaths [63]. This can be attributed to the significant attenuation of the hypothalamus–pituitary axis (HPA) response, suggesting a possible link between initial HPA-axis response disruption and subsequent unfavorable outcomes [64].

Symbicort: Steroids such as Symbicort exert potent anti-inflammatory properties but come with a spectrum of adverse effects from mild concerns like acne to severe conditions like Cushing syndrome, thus potentially leading to diabetes and heart problems [65]. They are widely employed to manage inflammatory and autoimmune diseases including rheumatoid arthritis, upper airway inflammation, asthma, and pulmonary conditions. However, corticosteroids can provoke psychiatric issues like depression, anxiety, delirium, and panic disorders [66]. The side effects impact up to 90% of long-term corticosteroid users and may even lead to cognitive impairment progressing to dementia or delirium. Multiple studies report varying rates of mental problems ranging from 2–60% [67]. However, dosage needs to be taken into consideration to avoid side effects. By discontinuing corticosteroids, one can resolve mood and cognition problems as demonstrated in multiple controlled experiments [68]. Studies investigating its risk in ASUD are lacking.

Indicators of decreased risk:

Clindamycin: Based on medical notes, we found that endocarditis, pneumonia, and osteomyelitis are popular indications for clindamycin in these PTSD patients. Postoperative and posttraumatic infections of bones and joints are some of the most common complications in the field of medicine. Osteomyelitis is a condition in which patients experience inflammation in the bone and bone marrow. This could be either due to tuberculosis or syphilis, or bacterial, fungal, or parasitic (toxoplasma gondii) in origin [69]. There is a growing interest in epidemiology where infections are implicated as a novel risk factor for the development of multiple mental disorders. The infection caused by the neurotropic parasite Toxoplasma gondii (T. gondii) is transmitted to a host (e.g., rodent or human) via the ingestion of tissue cysts in undercooked meat or oocytes in cat feces or contaminated soil, where it progresses to form focal brain lesions as seen in patients with acquired immune deficiency syndrome (AIDS) [70]. This can further lead to seizures, mental confusion, neurological impairment, ataxia, visual abnormalities, cranial nerve palsy, alcohol-related dementia, and psychomotor or behavioral alterations among patients with multiple etiologies [71]. A study examined the association between *T. gondii* infection, anxiety, PTSD, and depression among individuals in a population-based study. They found that seropositive individuals had more than twice the odds of reporting anxiety compared to seronegative individuals, suggesting a relationship between the immune response to *T. gondii* and other multiple anxiety and mood disorders [72]. A case study found that clindamycin treatment provided clinical improvement within 48 h of treatment and resolved irregular brain lesions in the right basal ganglia within 3 weeks of treatment [73]. Another case study proposed clindamycin to be used to improve the cognitive function of AIDS patients with cerebral toxoplasmosis and alcohol abuse [74,75,76]. Thus, the use of clindamycin can be extended to patients with PTSD and ASUD.

Recent studies have suggested no major improvement in community-acquired streptococcus pneumoniae meningitis. The long-term neurological sequelae coupled with its high mortality impacts overall quality of life. This can be attributed to the systemic inflammatory response of the host leading to leucocyte extravasation into the subarachnoid space, brain edema, secondary ischemia and vasculitis, stimulation of resident microglia in the central nervous system by bacterial compounds, and finally, interaction with the bacterial hemolysins on neurons [77,78,79]. A study performed in a rabbit model found that clindamycin was found to pass the blood–brain barrier and provided neuroprotection as opposed to other drugs in question. The proposed mechanism of action is attributed to reduced hydroxyl radical formation and lower concentrations of glutamate and glycerol in the interstitial fluid of the hippocampal formation, which finally leads to decreased neuronal injury in the dentate gyrus [80]. As mentioned, the role of the dentate gyrus in bipolar disorder [81], schizophrenia [82], and PTSD reiterates the possibility of clindamycin’s protective effects.

Enalapril: Disorders such as depression and anxiety manifest as excessive fear, hypervigilance, and related disturbances, often co-occurring with major depressive disorder (MDD). This interplay may be driven by heightened HPA axis activity and amygdala dysfunction. Conventional treatments often exhibit limited efficacy and delayed onset, exacerbated by accompanying anxiety, which underscores the urgent need to explore novel therapeutic targets. The renin-angiotensin system (RAS), traditionally associated with hypertension, has emerged as a potential player in these disorders. Elevated RAS activity is linked to depression and anxiety, partly due to neuroinflammation, stress, and oxidative stress induction. RAS blockade demonstrates anti-inflammatory and anti-oxidative stress properties, suggesting a foundation for treating depression and anxiety. Drugs like captopril and enalapril (angiotensin-converting enzyme inhibitors: ACEIs) have exhibited rapid mood improvement in hypertensive patients [83]. Another study found that ACEIs such as captopril and enalapril improved and reversed the adverse memory effects of hypertension. High arterial blood pressure is significantly associated with cognition impairment along with depression and anxiety [84]. Both reversed these deficits. Another study found that 64% of normotensive alcoholics taking 20 mg/day of enalapril had decreased their alcohol intake as compared to the control [85,86]. In turn, these inhibitors may reduce alcohol intake by elevating a nonapeptide fragment or elevating central angiotensin II levels [87].

Other medications such as penicillin, valacyclovir, Xarelto/rivaroxaban, moxifloxacin, diphenoxylate atropine, and sodium sulfacetamide sulfur could potentially serve as protective treatment options for PTSD patients with ASUD risks (See Appendix A for more details). It is worth noting that there may not be existing literature reports on these medications’ direct impact on ASUD risk, likely because their effects on ASUD risk are indirect. We also found that “bundled screening” for early detection and treatment of mental disorders, ASUDs, and other unspecified conditions could be used as a precautionary protective factor because it can help in overall healthcare cost reduction, disseminate complications from co-occurring disorders, and overcome the lack of adequate behavioral health infrastructure to provide appropriate diagnostic follow up. Other protective factors such as screening for malignant neoplasms of the cervix amongst PTSD and ASUD patients will help identify high-risk patients early on. Our results are in line with a study that applied the Health Belief Model and trauma-informed frameworks to guide their analysis. They found that discomfort with pap screening was common amongst women experiencing PTSD, ASUD, and homelessness and who had a history of sexual trauma such as interpersonal violence, incarceration, discrimination, and neglect [88]. Providers suggested an aggressive application of a trauma-informed approach where educating, counseling, and privacy may help address complex barriers among women experiencing PTSD, ASUD, and other discomforts.

### 4.3. Effect of SDoH on PTSD for ASUD Prediction

Negatively correlated:

Racial segregation: We found that patients who had values closer to −1 had higher incidences of ASUD risks as opposed to patients closer to 1. Our results are in line with a study that showed patients belonging to heavily racially segregated areas experiencing place-based health disparities, which often arise due to a result of historical segregationist policies, tend to have higher levels of inflammation that were attributed to higher incidences of mental disorders [89].

Income segregation: Like racial segregation, our findings indicate that patients with values closer to −1 are more prone to ASUD risks compared to those values closer to 1. These results align with a previous study that demonstrated how individuals residing in heavily segregated, low-income areas often face place-based health disparities. These disparities are frequently linked to historical segregationist policies and are associated with increased levels of inflammation, which, in turn, are connected to a higher occurrence of mental disorders [89].

Low Neighborhood Socioeconomic Index (nSES index): The neighborhood socio-economic index (nSES) captures the educational, occupational, and wealth composition of a given zip code, as well as the material resources available to the residents [90]. Neighborhood environments can impact health through several pathways, including psychosocial stress stemming from toxic social environments prevalent in socioeconomically deprived neighborhoods, individual-level factors such as smoking and diet, and exposure to toxic physical elements like air pollution and chemical pollutants, which are more common in low-income neighborhoods. These pathways may contribute to physiologic stress, with a focus on inflammation, known to lead to chronic diseases including cardiovascular disease, diabetes complications, and cancer [91]. Toxic social stressors can stimulate inflammatory responses via the HPA axis, resulting in chronic inflammation and increased risks. Higher levels of inflammatory markers in residents of low socioeconomic status neighborhoods support the notion that neighborhood deprivation may influence disease through inflammatory pathways [92]. We found that patients belonging to lower nSES zip codes had higher ASUD risks as opposed to those belonging to high nSES zip codes.

Younger patients: Traumatic events affecting young individuals at higher rates can have profound consequences on their mental health given the ongoing neurobiological and emotional development. Prevalence estimates for trauma exposure and PTSD in young individuals vary but underscore the need for updated assessments using current diagnostic criteria. These experiences are associated with a significant health burden, including increased risks of psychiatric disorders, suicidality, and functional impairment [93]. Furthermore, the United States is witnessing an alarming surge in ASUD among young individuals, emphasizing the need for healthcare professionals to adapt their perioperative care to minimize relapse rates in this demographic [94]. While further research is necessary, our results shed light on the possibility of younger individuals being susceptible to the addictive properties of various substances.

Other SDoH parameters, such as patients belonging to zip codes with a black majority, low normalized difference vegetation, low aridity, a high percentage of non-USA citizens, a high number of households with limited English-speaking capacity, a high number of widowed partners who are females, higher income segregation, and low Gini index, were all found to have an association with PTSD and ASUD, both in our study and in the mental health literature and, thus, should be considered when reforming the healthcare system to respond to the challenge of health disparities.

Positively correlated to ASUD risk:

Transportation barrier: Transportation barriers due to a lack of access to transportation lead to rescheduling conflicts, longer wait times, and missed or delayed care [95]. These, in turn, promote poorer management of chronic illness and mental health outcomes [96]. We found that patients living in zip codes with a lack of available transportation tend to experience higher incidences of ASUD risks as opposed to patients living in zip codes with access to transportation. Our study is in line with a study that showed that 5.8 million individuals in 2017 delayed medical care because they did not have access to transportation. The study emphasized that transportation barriers increased between 2003 and 2009 with people of color, those living below the poverty threshold, Medicaid recipients, and people with disabilities having greater odds of reporting a transportation barrier [97]. Another study found unwillingness to be in treatment, financial/insurance, and transportation barriers to be the most common barriers to aftercare treatment for alcohol and substance use [98].

Households with separated partners: The link between heavy drinking and divorce/separation has been recognized, with alcohol abusers having a roughly 20% higher risk of divorce compared to the general population [99]. However, little is known about the impact of divorce on the mental health of individuals in marriages marked by alcohol abuse. Current research on divorce in the general population has shown that mental health problems tend to peak following a divorce, possibly due to a combination of the health selection model (where troubled individuals are less likely to stay married) and the social causation model (stating that divorce-related adversities like emotional stress, unhealthy lifestyles, reduced social support, and limited resources lead to mental health issues) [100]. Our study is in line with another study that found that mental health may deteriorate more significantly after divorce among high-risk users, but that the impact may be less severe than in other divorced couples [101].

Cognitive behavioral therapy and Trauma-focused psychotherapy: Cognitive behavioral therapy serves as a psychotherapeutic approach aimed at identifying and transforming detrimental patterns that adversely impact one’s behavior and emotions. It proves particularly beneficial in helping individuals break free from pessimism and problem-solving challenges during times of stress, fostering more balanced thinking and enhancing their stress-coping abilities [102]. Also, trauma-focused psychotherapy such as cognitive processing therapy, prolonged exposure, and EMDR boast the strongest empirical support in addressing core PTSD symptoms. These techniques involve immersing individuals in prolonged and narrative exposure exercises, allowing them to reimagine and reframe traumatic experiences while restructuring cognitive processes. EMR entails revisiting distressing memories and their associated thoughts while simultaneously engaging in bilateral physical stimulation, such as eye movements, taps, or tones [103]. Research has identified these therapies as effective treatments for patients with PTSD [104]. While research using real-world data is lacking, our study is one of the first to find that PTSD patients undergoing these therapies had less risk of ASUD than those who did not undergo these treatments in a real-world setting.

Other SDoH parameters such as patients belonging to zip codes with households with same-sex marriages, single parents, and a higher number of patients who are white through EMR information were found to exhibit elevated incidences of PTSD, suicide-related events, and ASUD [95,105,106,107,108].

We also noticed that by adding the neighborhood level of SDoH parameters, the performance of Deepbiomarker2 has a slight improvement, and this might imply that some impact of those SDoH parameters has been captured by EMR data. As shown in Table 4, routine general medical examination at a healthcare facility, screening for malignant neoplasms of the cervix, and screening for childhood obesity are protective factors with reduced ASUD risk.

### 4.4. New Hypothesis Generated by DeepBiomarker2 on ASUD Risk in PTSD

Most prior PTSD and ASUD studies share a common shortcoming of being rooted in the de facto assumption that ASUDs may emerge majorly due to biological factors. However, we would like to propose some unique hypotheses that are based on our results and validated by the literature. They may serve as important novel interdisciplinary indicators of mental health diseases. However, it is important to approach our findings with caution since they do not establish a causal relationship. There exists a possible indirect link between gut microbiota dysbiosis, UTIs, and Toxoplasmosis gondii [109]. Microbial dysbiosis is a major perpetrator of intestinal inflammation, which promotes subsequent permeability of the gut barrier, ultimately leading to distal consequences of Toxoplasmosis gondii infection to permeate the blood–brain barrier [110]. This compromise may promote cognition and AUD risk in patients with PTSD, depression, epilepsy, suicidal ideation, GAD, and schizophrenia [111]. This health disparity can be potentially corrected by improved water purification for all. Serum albumin (SA) is closely related to oxidative stress and antioxidant capacity. It may exist in relatively high concentrations in patients with liver disease and other neurodegeneration disorders, and significantly lower in patients with cancer, critically ill patients, and patients with neuropsychiatric disorders such as schizophrenia and depression [112]. Albumin may aggravate oxidative stress by increasing the percentage of free radicals and oxidative damage products entering the blood–brain barrier. This, in turn, may promote inflammation while simultaneously decreasing omega-3 polyunsaturated fatty acids, magnesium, and thyroid hormone levels causing depression [113,114,115,116]. Another unconventional hypothesis that could be extrapolated and used for future studies is correlating allergies to PTSD and ASUD. A study showed immunoglobulin-E and WBCs to be associated with worsening depressive scores in bipolar patients during high pollen seasons. Also, PTSD patients with nicotine dependence and chronic obstructive pulmonary disorder (COPD) had high RBCs [117]. Based on our results and the literature, we show that the activation of inflammatory mediators due to asthma and allergy rhinitis may be a potential biomarker for predicting ASUDs in PTSD, but further investigations are necessary.

### 4.5. Biomarkers for Personalized Treatment for PTSD to Reduce the Risk of ASUD

Using EMR and non-EMR data, we examined multiple biomarkers concerning patient medication history, diagnosis, lab tests, SDoH (both neighborhood and patient levels), and psychotherapy to propose novel therapies for PTSD patients with ASUDs. Our findings provided compelling evidence that suggests that in addition to targeting risk and protective factors and developing prevention and intervention strategies, one must incorporate SDoH and psychotherapy information to ameliorate sources of psychological and psychosocial risk. This, in turn, would help to better predict ASUD risks in PTSD patients. Although the literature focuses on using conventional SDoH parameters such as income, race, and unemployment to be the major predicting factors for future ASUDs, the inclusion of future SDoH-ASUD factors such as peer substance use, easy drug access, cultural norms, low access to ASUD help centers, and other psychosocial parameters should be considered to obtain a comprehensive view of PTSD and SDoH. Our tool is intuitive and could provide easy interpretation of these complex biomarkers. We emphasize a novel focus on biomarkers for assessing ASUD risk among PTSD patients, particularly the inclusion and examination of SDoH and psychotherapy parameters. This pioneering approach identifies novel biomarkers that are relevant to both PTSD and ASUD, a distinctive contribution considering previous studies focused solely on either condition. Additionally, our model’s ability to integrate diverse factors, including SDoH and psychotherapy, sets it apart from prior research. We emphasize the significance of considering combinations of biomarkers over individual ones. Our study’s robustness encompasses a wide range of both EMR (both clinically and non-clinically applicable) biomarkers and non-EMR (non-medical and non-behavioral precursors of health) biomarkers, features a substantial sample size, considers sequential effect, and acts as an efficient choice of routine testing, which positions it as a valuable contribution to the existing literature and provides a more comprehensive understanding of these complex conditions. Our study’s emphasis on refining our relative contribution analysis adds a valuable dimension to the existing literature and enhances our understanding of the identification and treatment of ASUD among PTSD patients. While it is true that many of the factors associated with PTSD and ASUD are well-established in prior research, our analysis is the first to provide several noteworthy contributions concerning both PTSD and ASUD that deserve attention. Firstly, our approach allows for the replication of previous findings, which indirectly validates the effectiveness of our analytical methodology. By confirming established non-causal associations between certain factors and comorbid PTSD and ASUD, we demonstrate the reliability and robustness of our approach. This validation serves as a foundation upon which we build our novel findings. While certain biomarkers may be well-known in isolation, our study uniquely connects the biomarkers associated with both PTSD and ASUD. This linkage is a distinctive contribution, as it underscores the interplay between these conditions and offers fresh insights into potential diagnostic and therapeutic avenues. Moreover, the extensive literature we have cited serves to establish the context and significance of our work. It demonstrates that while individual factors have been explored, the comprehensive analysis of ASUD and PTSD biomarkers is relatively underexplored. Our study bridges this gap by systematically examining a wide range of factors within the context of comorbid PTSD and ASUD and provides a more comprehensive perspective on the identification and treatment of ASUD among PTSD patients.

Our study also has a few limitations: First, there could be inconsistencies in biochemical test results between patients due to enrollment bias, and some lab tests might have low representation in our database. As such, the analysis might have limited power to detect the effects. Second, we used EMR data from January 2004 to October 2020, and in this period, there is a possibility of changes in treatment and number of lab tests amongst these patients. However, these are limitations caused by using observational EMR as a data source and can be resolved by investigations using a randomized clinical trial or prospective design. Third, we considered the effect of biomarkers along with diagnosis and medication use; however, in our results, comorbidities had a higher impact compared to biomarkers. This can be explained by the fact that diagnosis considers the past status of the patients while biomarkers take into consideration the recent status of PTSD and ASUDs. However, it is important to note that these associations are not causal, and further investigation is planned to explore potential causal relationships involving these biomarkers. Fourth, there are SDoH data inconsistencies due to missing data. Fifth, due to data limitations, we only mapped neighborhood-level SDoH parameters based on the zip codes of patient information extracted from EMR data, and we did not include individual-level SDoH (e.g., income, home address, and education level) due to patient protection and privacy issues. Sixth, while our primary findings solidly establish predictive features for ASUD risk in PTSD, caution should be exercised when interpreting the secondary results related to specific features. This caution is particularly relevant in the context of deep learning models as these methods assess factors/features independently, whereas the original predictive models inherently consider combinations of factors/features. This challenge in interpretation is inherent to such models and extends beyond the scope of our paper. Seventh, our use of diagnosis codes to identify ASUD has inherent limitations, particularly concerning questions related to indicators of increased and decreased risk. We did not include a formal diagnosis by conducting screening interviews to formally diagnose these patients but used the diagnosis codes available to us from published results [118]. It is plausible that the observed impact of diagnosed comorbidities may reflect variations in detection and documentation practices rather than solely underlying disease processes. The diagnosis codes are simply extracted from claims data and are not a representation of the actual diagnosis of the patient, which is a limitation of using EMR data. Eighth, we used SDoH measures at the 5-digit zip code level, which may result in reduced precision compared to finer geographical units. However, prior research supports the use of 5-digit zip code-level data as a practical approach for examining SDoH, allowing us to contribute valuable insights into its relationship with the ASUD risk [37]. Because of the inclusion of SDoH in our model, it is hard to compare our model performance with other approaches because those models did not integrate SDoH yet. Instead, we used evidence from the literature to validate our model. There might still be room to enhance the model performance with more recent models. In addition, although transformer models exhibit superior performances, their implementation in our study proves challenging due to the constraints imposed by our small dataset [18]. Our future research will seek to find remedies to the above shortcomings by utilizing larger and more integrated datasets with more accurate diagnoses, improving our algorithms, including more insightful biomarkers, integrating other forms of metadata and other biological measures relevant to ASUD risk, and performing causal pathway analyses.

## 5. Conclusions

Our improvised data-driven deep learning approach aimed to identify and examine biomarkers for assessing the risk of ASUDs among PTSD patients, which can be utilized to develop novel interdisciplinary hypotheses surrounding its etiology. Extrapolating our results and current information, we found medications like clindamycin, enalapril, penicillin, valacyclovir, Xarelto/rivaroxaban, moxifloxacin, sodium sulfacetamide sulfur, diphenoxylate, and atropine all to have a potential to reduce risk of ASUDs among PTSD patients. That, being said we also found multiple SDoH parameters, both conventional and unconventional ones, and psychotherapy to have significant contributions to ASUD risk prediction. The high accuracy of DeepBiomarker2 showed ASUD risk to be particularly higher among a subset of patients who may experience elevated psychosocial risk coupled with pain medication treatment and inactive psychotherapy status. This might provide a novel insight into our understanding of PTSD with ASUD in a more holistic manner. While universal prevention programs may offer current benefits, these findings from DeepBiomarker2 offer potentially valuable and refined information that can be used to design and develop personalized prevention and intervention programs that are designed to address the psychosocial needs and health disparities existing amongst these high-risk patients.

## Figures and Tables

**Figure 1 jpm-14-00094-f001:**
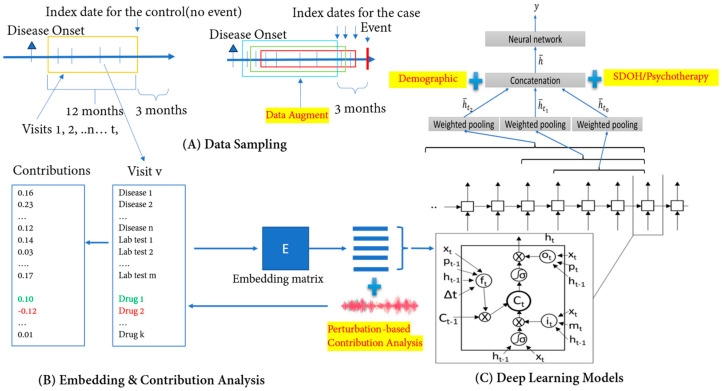
Overview of DeepBiomarker2. (**A**) Data sampling from electronic medical records: Patients A and B both pass the inclusion criteria and within the given time interval. Patient A has no event and Patient B has events and is considered as a control and a case, respectively. We extract their multimodal information (i.e., Diagnoses/Disease, Medication use/Drug, and Lab-test results) from their EMRs and use them as input in our model; (**B**) data embedding: the multimodal information is then converted into continuous vector spaces to build an embedding matrix; and (**C**) prediction by neural networks such as TLSTM and RETAIN as the basic prediction units. We then incorporate individual- and neighborhood-level SDoH information, trauma-focused psychotherapy, and cognitive behavioral status information in our neural networks for outcome prediction. Our model provides us with a comprehensive list of multiple biomarkers, and with the help of the perturbation-based contribution analysis, we identify the most important features/biomarkers. TLSTM: Time-aware long short-term memory; RETAIN: Reverse Time Attention Model; SDoH: social determinants of health.

**Table 1 jpm-14-00094-t001:** The performance metrics of DeepBiomarker2 with different deep-learning and machine-learning algorithms with and without SDoH features.

**RETAIN(+SDOH)**	**1**	**2**	**3**	**4**	**5**	**Average**	**std.s**
**Validation AUC**	0.922	0.925	0.927	0.934	0.935	0.929	0.005
**Test AUC**	0.918	0.927	0.930	0.933	0.928	0.927	0.007
**Test Precision**	0.886	0.890	0.899	0.915	0.886	0.895	0.013
**Test Recall**	0.850	0.878	0.865	0.867	0.869	0.866	0.011
**Test F1**	0.868	0.884	0.882	0.890	0.877	0.880	0.009
**RETAIN(−SDOH)**	**1**	**2**	**3**	**4**	**5**	**Average**	**std.s**
**Validation AUC**	0.926	0.921	0.933	0.933	0.922	0.927	0.006
**Test AUC**	0.923	0.918	0.916	0.932	0.927	0.923	0.006
**Test Precision**	0.898	0.878	0.857	0.888	0.866	0.878	0.017
**Test Recall**	0.867	0.870	0.872	0.882	0.881	0.874	0.007
**Test F1**	0.882	0.874	0.864	0.885	0.873	0.876	0.008
**LR(+SDOH)**	**1**	**2**	**3**	**4**	**5**	**Average**	**std.s**
**Validation AUC**	0.875	0.872	0.871	0.876	0.868	0.872	0.003
**Test AUC**	0.849	0.841	0.845	0.854	0.844	0.847	0.005
**Test Precision**	0.758	0.702	0.714	0.746	0.723	0.729	0.023
**Test Recall**	0.825	0.872	0.858	0.849	0.838	0.848	0.018
**Test F1**	0.790	0.778	0.779	0.794	0.776	0.784	0.008
**LR(-SDOH)**	**1**	**2**	**3**	**4**	**5**	**Average**	**std.s**
**Validation AUC**	0.869	0.868	0.869	0.851	0.868	0.865	0.008
**Test AUC**	0.846	0.843	0.843	0.828	0.841	0.840	0.007
**Test Precision**	0.704	0.706	0.692	0.752	0.773	0.725	0.035
**Test Recall**	0.897	0.889	0.904	0.776	0.787	0.851	0.063
**Test F1**	0.789	0.787	0.784	0.764	0.780	0.781	0.010
**TLSTM(+SDOH)**	**1**	**2**	**3**	**4**	**5**	**Average**	**std.s**
**Validation AUC**	0.936	0.932	0.935	0.944	0.929	0.935	0.006
**Test AUC**	0.910	0.924	0.935	0.928	0.935	0.926	0.010
**Test Precision**	0.786	0.863	0.869	0.873	0.886	0.855	0.040
**Test Recall**	0.906	0.855	0.887	0.855	0.878	0.876	0.022
**Test F1**	0.842	0.859	0.878	0.864	0.882	0.865	0.016
**TLSTM** **(−SDOH)**	**1**	**2**	**3**	**4**	**5**	**Average**	**std.s**
**Validation AUC**	0.931	0.939	0.937	0.937	0.941	0.937	0.004
**Test AUC**	0.919	0.925	0.925	0.923	0.931	0.924	0.005
**Test Precision**	0.826	0.869	0.869	0.836	0.882	0.857	0.024
**Test Recall**	0.902	0.878	0.859	0.888	0.859	0.877	0.019
**Test F1**	0.862	0.874	0.864	0.862	0.871	0.866	0.005

AUC: area under curve, std: standard deviation, TLSTM: Time-Aware Long Short-Term Memory, RETAIN: Reverse Time AttentIoN model, LR: Logistic regression, std.s: Standard deviations of validation AUC, test AUC, test precision, test recall, and test F1, respectively.

**Table 2 jpm-14-00094-t002:** Most important abnormal lab test results identified by perturbation-based contribution analysis for ASUD prediction.

**Feature Name**	**Relative Contribution**	**Wilcoxon_p**	**FDR_Q**
HGB	1.46	2.63 × 10^−34^	8.75 × 10^−35^
HCT	1.40	5.43 × 10^−29^	1.36 × 10^−29^
Glucose	1.32	9.64 × 10^−28^	1.93 × 10^−28^
RBC	1.28	4.99 × 10^−16^	4.99 × 10^−17^
WBC	1.32	6.02 × 10^−14^	5.47 × 10^−15^
CL	1.29	8.99 × 10^−14^	7.49 × 10^−15^
MCHC	1.34	9.79 × 10^−13^	7.53 × 10^−14^
MCH	1.31	4.18 × 10^−11^	2.78 × 10^−12^
Albumin	1.35	5.77 × 10^−11^	3.61 × 10^−12^
RDW	1.30	7.80 × 10^−11^	4.59 × 10^−12^
Total Protein	1.41	1.31 × 10^−9^	6.57 × 10^−11^
Protein-Urine	1.41	2.20 × 10^−9^	1.00 × 10^−10^
Leukocyte Esterase	1.36	9.21 × 10^−9^	3.54 × 10^−10^
ABS Neutrophils	1.33	1.95 × 10^−8^	6.95 × 10^−10^
CO_2_	1.28	0.0001	2.87 × 10^−6^
Urea Nitrogen	1.20	0.0001	2.87 × 10^−6^
Ca	1.27	0.0003	5.77 × 10^−6^

Relative contribution value >1: Risk and Relative contribution value, <1: FDR_Q: false discovery rate adjusted Q value, p_wilcoxon: *p* values of Wilcoxon test. Hemoglobin (HGB), hematocrit (HCT), red cell distribution width (RDW), red blood cells (RBC), white blood cells (WBC), absolute (ABS) neutrophils, mean corpuscular hemoglobin (MCH), mean corpuscular hemoglobin concentration (MCHC), chloride (CL), carbon dioxide (CO_2_), and calcium (Ca).

**Table 3 jpm-14-00094-t003:** Most important medication uses results identified by perturbation-based contribution analysis for ASUD prediction.

Feature Name	Relative Contribution	Wilcoxon_p	FDR_Q
Acetaminophen	1.60	3.67 × 10^−38^	1.84 × 10^−38^
Hydrocodone	1.42	1.38 × 10^−10^	7.66 × 10^−12^
Oxycodone	1.44	3.49 × 10^−10^	1.84 × 10^−11^
Diphenoxylate and Atropine	0.30	4.07 × 10^−8^	1.40 × 10^−9^
Sodium sulfacetamide sulfur	0.45	2.52 × 10^−7^	7.87 × 10^−9^
Gabapentin	1.39	4.16 × 10^−7^	1.26 × 10^−8^
Enalapril	0.48	0.0001	2.84 × 10^−6^
Moxifloxacin	0.31	0.0003	5.46 × 10^−6^
Alprazolam	1.63	0.0019	2.89 × 10^−5^
Symbicort	1.36	0.0228	0.0003
Clindamycin	0.59	0.0295	0.0003
Xarelto/Rivaroxaban	0.39	0.0337	0.0004
Valacyclovir	0.53	0.0495	0.0005
Penicillin	0.65	0.0735	0.0007

Relative contribution value >1: Risk and Relative contribution value, <1: FDR_Q: false discovery rate adjusted Q value, p_wilcoxon: *p* values of Wilcoxon test.

**Table 4 jpm-14-00094-t004:** Most important diagnosis results identified by perturbation-based contribution analysis for ASUD prediction.

**Feature Name**	**Relative Contribution**	**Wilcoxon_p**	**FDR_Q**
Routine general medical examination at a health care facility	0.71	9.76 × 10^−25^	1.63 × 10^−25^
Esophageal reflux	1.25	1.40 × 10^−16^	1.68 × 10^−17^
Asthma, unspecified type, unspecified	1.34	1.51 × 10^−16^	1.68 × 10^−17^
Long-term (current) use of anticoagulants	1.24	4.82 × 10^−9^	2.10 × 10^−10^
Personal history of tobacco use	1.25	8.78 × 10^−9^	3.51 × 10^−10^
Anxiety state, unspecified	1.18	1.21 × 10^−8^	4.49 × 10^−10^
Lumbago	1.25	8.80 × 10^−7^	2.45 × 10^−8^
Personal history of other mental disorders	1.28	1.15 × 10^−6^	3.10 × 10^−8^
Depressive disorder, not elsewhere classified	1.21	4.54 × 10^−6^	1.16 × 10^−7^
Periumbilical pain	1.17	1.93 × 10^−5^	4.48 × 10^−7^
Fibromyalgia	1.19	0.0024	3.58 × 10^−5^
Migraine without aura, with intractable migraine, so stated, without mention of status migrainosus	1.17	0.0146	0.0002
Osteoarthrosis, unspecified whether generalized or localized, site unspecified	1.26	0.0157	0.0002
Arthrodesis status	1.18	0.0192	0.0002
Body Mass Index, pediatric, greater than or equal to 95th percentile for age	0.86	0.0261	0.0003
Other chronic pain	1.17	0.0704	0.0007
Screening for malignant neoplasms of cervix	0.86	0.5997	0.0037

Relative contribution value >1: Risk and Relative contribution value, <1: FDR_Q: false discovery rate adjusted Q value, p_wilcoxon: *p* values of Wilcoxon test.

**Table 5 jpm-14-00094-t005:** Most important SDoH and psychotherapy parameters identified by averaging 10 repeats for ASUD prediction.

Name	Mean	sd	*p*	Impact on ASUD Risk	Type of SDoH
Race (White)	0.120	0.02	2.60 × 10^−9^	White patients have higher risk of ASUD	Individual
Trauma focused psychotherapy	−0.0812	0.016	1.49 × 10^−8^	Individuals undergoing trauma focused psychotherapy have lesser risk of ASUD	Individual
Neighborhood socioeconomic status	−0.0943	0.033	3.27 × 10^−6^	Neighborhoods with low socio-economic status has higher risk of ASUD	Neighborhood
Percentage of Non-Citizens	−0.116	0.04	3.43 × 10^−6^	Non-US Citizens have a higher chance of ASUD risk	Neighborhood
Cognitive behavioral therapy	−0.0623	0.022	3.66 × 10^−6^	Individuals undergoing cognitive behavioral therapy have lesser risk of ASUD	Individual
Percentage of Foreign born	−0.0901	0.033	5.08 × 10^−6^	US born patients have higher risk of ASUD	Neighborhood
People of color index	−0.0778	0.03	9.88 × 10^−6^	Black majority have higher risk of ASUD	Neighborhood
Limited English-speaking household	−0.106	0.042	1.00 × 10^−5^	Households with limited English speaking capacity have higher risk of ASUD	Neighborhood
Racial segregation	−0.118	0.047	1.30 × 10^−5^	High racially segregated zip codes have higher risk of ASUD	Neighborhood
Widowed partner who is a Male	−0.0781	0.032	1.41 × 10^−5^	Widowed partner who is a male have lower risk of ASUD as opposed to widowed partner who is a female	Neighborhood
Household with transportation barriers	0.0870	0.04	3.82 × 10^−5^	Households with no vehicles have higher risk of ASUD	Neighborhood
Gender	−0.0964	0.046	5.56 × 10^−5^	Females have higher risk of ASUD	Individual
Age	−0.0828	0.043	0.0001	Younger patients have higher risk of ASUD	Individual
Household with same sex marriages	0.0652	0.039	0.0003	Households with same sex marriages have a higher chance of ASUD risk	Neighborhood
Aridity	−0.0487	0.03	0.0004	Low Humidity/lower vegetation/greenery have higher risk of ASUD	Neighborhood
Normalized difference vegetative index	−0.0581	0.043	0.0016	Low vegetation/greenery have higher risk of ASUD	Neighborhood
Gini index	−0.0216	0.019	0.0045	Zip codes with low Gini index have higher risk of ASUD	Neighborhood
Household with Separated partners	0.0396	0.037	0.0063	Households with separated partners have higher risk of ASUD	Neighborhood
Income segregation	−0.0276	0.036	0.0309	Households with higher income segregation have higher risk of ASUD	Neighborhood

SDoH: social determinants of health, sd: standard deviation, *p*: *p* values.

## Data Availability

The data used in this study were from UPMC under a data use agreement. The authors are not permitted to distribute the data to any third party, but researchers may contact UPMC for data access.

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
