# Peer review of "DeepBiomarker2: Prediction of Alcohol and Substance Use Disorder Risk in Post-Traumatic Stress Disorder Patients Using Electronic Medical Records and Multiple Social Determinants of Health"

_jpm, 2024, doi:10.3390/jpm14010094_

Round 1
Reviewer 1 Report
Comments and Suggestions for Authors
The author proposed a prediction technique for alcohol and substance use disorder risk using deep learning techniques. My comments about this work are given below:
- Remove the following statement from the abstract:
“We improved our previous deep learning model, DeepBiomarker, to develop DeepBiomarker2 through data integration of multimodal information from electronic medical records (EMR) data.”
- Discuss only the proposed work in the abstract section.
- Highlight the novelty of the research in the abstract.
- Mention the performance metrics values in the abstract instead of mentioning high accuracy like general terms.
- Summarize the limitations of the existing approaches at the end of Section 1 (Introduction).
- It is recommended to create a literature survey section for discussion about the existing related works and their limitations.
- Use visualization techniques to show the sample data or properties.
- Training and validation plots for the deep learning models are missing.
- The result section needs to be improved with several validation techniques. Proper justification is needed for how the proposed work performs better than the existing work.
- The advantages and limitations of the proposed work need to be clearly explained in the conclusion section.
Author Response
We would like to thank the reviewers for the constructive feedback and would like to discuss in detail the changes made with respect to the manuscript. We would like to mention that we have included the limitations pointed out in our current version of the manuscript. We answered the feedback point by point. Please see below:
Reviewer 1.
The author proposed a prediction technique for alcohol and substance use disorder risk using deep learning techniques. My comments about this work are given below:
- Remove the following statement from the abstract:
“We improved our previous deep learning model, DeepBiomarker, to develop DeepBiomarker2 through data integration of multimodal information from electronic medical records (EMR) data.”
Response: As suggested by the reviewer, we removed this sentence from the abstract.
- Discuss only the proposed work in the abstract section.
Response: As suggested by the reviewer, we have only included the proposed work in the abstract.
- Highlight the novelty of the research in the abstract.
Response: We now highlighted the novelty of this research in the abstract: integration of SDoH and the refined contribution analysis to increase model interpretability.
- Mention the performance metrics values in the abstract instead of mentioning high accuracy like general terms.
Response: Thanks for your suggestions. We now added the performance metrics in the abstract.
- Summarize the limitations of the existing approaches at the end of Section 1 (Introduction).
Response: Thank you for the suggestions. We have summarized the limitations of the current deep learning approaches in the introduction section.
- It is recommended to create a literature survey section for discussion about the existing related works and their limitations.
Response: Thank you for the suggestions. We have also included a paragraph in the introduction section discussing the existing deep learning models in the field.
- Use visualization techniques to show the sample data or properties
Response: Thank you for the suggestions. We have included a table for baseline comparison in the supplementary Table 1.
- Training and validation plots for the deep learning models are missing.
Response: Thank you for the suggestions. We have included an example training and validation plot in the supplementary Figure 3.
- The result section needs to be improved with several validation techniques. Proper justification is needed for how the proposed work performs better than the existing work.
Response: We validate our clinical findings through an extensive literature search. The incorporation of SDoH into our model introduces a unique framework, making direct comparisons with existing approaches challenging. Our analysis involves established methodologies, including TLSTM and RETAIN. Their performances have been previously assessed by others. Additionally, we compared RETAIN, TLSTM and LR algorithms in our study.
- The advantages and limitations of the proposed work need to be clearly explained in the conclusion section.
Response: Thank you for comments. We here emphasize the advantage of including SDoH in our model and model interpretability by the refined perturbation-based approach in the conclusion section. We have mentioned that transformer models, as highlighted in the specific paper, demonstrate superior performance. However, due to the constraints of a small dataset like ours, constructing a transformer model presents a significant challenge for our study. As mentioned above in response to comment 9, the direct comparison between our method with others is challenging because of inclusion of the SDoH component in our model.
Reviewer 2 Report
Comments and Suggestions for Authors
The authors integrated electronic medical records with social determinants of health to predict alcohol and substance use disorder risk in PTSD patients, utilizing deep learning and natural language processing for data analysis from a large sample (38,807 patients).in my opinions, the findings have clinical implications, potentially aiding in the early identification and intervention for at-risk PTSD patients. My comments are as follosw:
1. In the abstract and some places, there are trunciated words like: Univer-sity , pre-dicted, etc. please check.
2. why 8:1:1, I believe many use 6:2:2?
3. "We leveraged the Pytorch_EHR framework developed by Zhi Group as the founda-198 tion of our analysis. These deep learning models based on multiple recurrent neural net-199 works, were used to analyze and predict clinical outcomes [24].
It's not proper to say "developed by Zhi Group". Please do the right thing by citing first author XXX. et. al.
4. While the paper utilizes advanced deep learning methods, there could be a greater focus on making the model more interpretable for clinical applications. Please comment on the model interpretability.
5. The model predicts ASUD risk within a three-month window. Extending this to longer-term predictions could enhance its utility. Have you tried for longer-term Predictive Power?
6. I am a bit confused how the study is ablE to lead to the medication. From Figure 1 and the main text, there is no mention about this. Kindly, elaborate and explain.
7. Can the authors make open access the dataset (de-identified) they have used? it is not found in the SI or appendix.
Overall, the manuscript can only be accepted after the above comments are addressed.
Comments on the Quality of English Language-
Author Response
We would like to thank the reviewers for the constructive feedback and would like to discuss in detail the changes made with respect to the manuscript. We would like to mention that we have included the limitations pointed out in our current version of the manuscript. We answered the feedback point by point. Please see below:
Reviewer 2
The authors integrated electronic medical records with social determinants of health to predict alcohol and substance use disorder risk in PTSD patients, utilizing deep learning and natural language processing for data analysis from a large sample (38,807 patients).in my opinions, the findings have clinical implications, potentially aiding in the early identification and intervention for at-risk PTSD patients. My comments are as follow:
- In the abstract and some places, there are truncated words like: Univer-sity , pre-dicted, etc. please check.
Response: Thank you for letting us know. The problem has been fixed.
- why 8:1:1, I believe many use 6:2:2?
Response: A recent article available at https://medium.com/@itsmeSamrat/splitting-the-data-to-60-20-20-ratio-vs-80-10-10-which-one-is-better-bbc3503830d8 has compared two ratios and concluded that 8:1:1 is more favorable. As the author rightly pointed out, the choice of ratio should be tailored to the specific characteristics of the datasets and “can be adjusted based on the specific requirements of the project.” In our scenario, where the dataset is relatively small, our objective is to maximize data utilization for training the model, aiming to produce high-performance models for feature analysis.
- "We leveraged the Pytorch_EHR framework developed by Zhi Group as the founda-198 tion of our analysis. These deep learning models based on multiple recurrent neural net-199 works, were used to analyze and predict clinical outcomes [24].
It's not proper to say "developed by Zhi Group". Please do the right thing by citing first author XXX. et. al.
Response: Thank you for your suggestion. We have rephrased the sentence to make it more professional and appropriate.
- While the paper utilizes advanced deep learning methods, there could be a greater focus on making the model more interpretable for clinical applications. Please comment on the model interpretability.
Response: Thank you for the comments on model interpretability. We have included the following statements explaining how the model is interpretable:
We examined the importance of the biomarkers/features for the prediction of ASUDs, we calculated the relative contribution (RC) of each biomarker/feature on ASUD. The RC of a biomarker/feature was calculated as the median contribution of the biomarkers/features to events divided by the median contributions of this c to no events. The contributions were estimated by a perturbation-based approach. The RC value and significance calculation are shown as follows where FC represents the feature contribution:
We have published the previous iteration of our model where we evaluated our approach using previous well-known knowledge to validate the performance of the analysis, find new biomarkers/features (e.g. Diagnoses/Disease, Medication use/Drug, Lab-test results, SDoH parameters) and to reveal some new insights for hypothesis generation. Our model incorporated clinical domain knowledge extracted from EMR data and involves collaborations with healthcare professionals, addressing a gap observed in other existing models. As mentioned in the newly added literature review, model interpretability is one of the current foci of deep learning field. Our approach can contribute to understanding the “black box” under the complex deep learning models.
- The model predicts ASUD risk within a three-month window. Extending this to longer-term predictions could enhance its utility. Have you tried for longer-term Predictive Power?
Response: Thank you for the question. Yes, we acknowledge the potential value of extending the predictive window for ASUD risk beyond three months. We experimented with a 6-month time window and observed very similar performance. The rationale behind selecting a 3-mmonth prediction time window and a one-year observation time window is rooted in our interest in predicting near-future risks (within 3 months) and understanding the impacts of medication use. Our assumption is based on the belief that the effects of a medication are not likely to endure for an extended period, such extending one year.
- I am a bit confused how the study is ablE to lead to the medication. From Figure 1 and the main text, there is no mention about this. Kindly, elaborate and explain.
Response: Our objective was to utilize multimodal information as biomarkers/features for a comprehensive understanding of PTSD and ASUD. We incorporated structured (e.g., Diagnoses/Disease, Medication/Drug, Lab-test results), unstructured (e.g., psychotherapy and veteran status) EMR, and open-source data (e.g., SDoH parameters). The input data consisted of the multimodal information from the year preceding the index date. Medication names were converted to unique DrugBank IDs and organized into a sequence. Using perturbation-based contribution analysis, we determined the biomarkers/features (e.g., medication, lab tests, diagnoses and SDoH) present in PTSD patients with high and low ASUD risk. The relative contribution (RC) of each biomarker/feature to ASUD prediction was calculated by comparing contributions with and without events. Normalized FC values and scaled RC values were used for all biomarkers/features. An RC value greater than 1 indicated a risk factor, while less than 1 indicated a beneficial factor. Our model, validated in a previous iteration, demonstrated performance, identified new biomarkers/features, and provided insights for hypothesis generation. As shown in Figure 1, we can identify medications(drugs) with RC values larger or smaller than 1, and those drugs can be further studied for their effects on PTSD and ASUD.
- Can the authors make open access the dataset (de-identified) they have used? it is not found in the SI or appendix.
Response: Unfortunately, we are unable to provide open access to the de-identified dataset that was utilized in this paper because of our data use agreement with UPMC. We have also attached the Data Use Agreement for your reference.
Round 2
Reviewer 1 Report
Comments and Suggestions for Authors
The authors have significantly improved the content in the current version. I recommend accepting the present form.
Reviewer 2 Report
Comments and Suggestions for Authors
The revised manuscript can now be accepted.
Comments on the Quality of English Language-